# Chlorogenic Acid Decreases Glutamate Release from Rat Cortical Nerve Terminals by P/Q-Type Ca^2+^ Channel Suppression: A Possible Neuroprotective Mechanism

**DOI:** 10.3390/ijms222111447

**Published:** 2021-10-23

**Authors:** Yi-Chieh Hung, Yi-Hsiu Kuo, Pei-Wen Hsieh, Ting-Yang Hsieh, Jinn-Rung Kuo, Su-Jane Wang

**Affiliations:** 1Chi Mei Medical Center, Department of Neurosurgery, Tainan 71004, Taiwan; hungyichieh@gmail.com (Y.-C.H.); koujinnrung@gmail.com (J.-R.K.); 2Graduate Institute of Biomedical and Pharmaceutical Science, Fu Jen Catholic University, New Taipei City 24205, Taiwan; maru.a3m@gmail.com; 3Research Center for Chinese Herbal Medicine, College of Human Ecology, Chang Gung University of Science and Technology, Taoyuan 33303, Taiwan; pewehs@mail.cgu.edu.tw; 4Graduate Institute of Natural Products, School of Traditional Chinese Medicine, Graduate Institute of Biomedical Sciences, College of Medicine, Chang Gung University, Taoyuan 33303, Taiwan; 5Program in Nutrition & Food Science, Fu Jen Catholic University, New Taipei City 24205, Taiwan; david711young@gmail.com; 6School of Medicine, Fu Jen Catholic University, New Taipei City 24205, Taiwan

**Keywords:** chlorogenic acid, glutamate release, P/Q-type Ca^2+^ channel, CaMKII, neuroprotection, cerebral cortex, synaptosome, kainic acid

## Abstract

The glutamatergic neurotransmitter system has received substantial attention in research on the pathophysiology and treatment of neurological disorders. The study investigated the effect of the polyphenolic compound chlorogenic acid (CGA) on glutamate release in rat cerebrocortical nerve terminals (synaptosomes). CGA inhibited 4-aminopyridine (4-AP)-induced glutamate release from synaptosomes. This inhibition was prevented in the absence of extracellular Ca^2+^ and was associated with the inhibition of 4-AP-induced elevation of Ca^2+^ but was not attributed to changes in synaptosomal membrane potential. In line with evidence observed through molecular docking, CGA did not inhibit glutamate release in the presence of P/Q-type Ca^2+^ channel inhibitors; therefore, CGA-induced inhibition of glutamate release may be mediated by P/Q-type Ca^2+^ channels. CGA-induced inhibition of glutamate release was also diminished by the calmodulin and Ca^2+^/calmodilin-dependent kinase II (CaMKII) inhibitors, and CGA reduced the phosphorylation of CaMKII and its substrate, synapsin I. Furthermore, pretreatment with intraperitoneal CGA injection attenuated the glutamate increment and neuronal damage in the rat cortex that were induced by kainic acid administration. These results indicate that CGA inhibits glutamate release from cortical synaptosomes by suppressing P/Q-type Ca^2+^ channels and CaMKII/synapsin I pathways, thereby preventing excitotoxic damage to cortical neurons.

## 1. Introduction

Glutamate is one of the most abundant excitatory neurotransmitters and performs vital physiological functions related to synaptic plasticity, memory, and learning [1,2]. In pathological conditions such as brain ischemia, seizures, and neurodegeneration, however, glutamate is released excessively, causing the over-activation of glutamate receptors. This process increases the concentrations of intracellular Ca^2+^, which can lead to neuronal damage or death [3,4]. The inhibition of glutamate release could serve as a useful therapeutic strategy for neuroprotection. Therefore, compounds capable of inhibiting glutamate release are promising for the development of neuroprotective drugs [5,6,7].

Many studies have explored the effects and applications of bioactive compounds, especially polyphenols, because of their benefits to human health, including reduction of inflammation, prevention of cancer, and promotion of cognition [8,9]. Polyphenols, which are commonly found in tea, coffee, and many fruits and vegetables, have been shown to inhibit the release of glutamate, thereby protecting neurons [10,11,12]. Chlorogenic acid (CGA), one of the most common polyphenols, has various biological benefits, including antioxidant, anti-inflammatory, antimicrobial, antihypertensive, and anticarcinogenic functions [13,14]. CGA can cross the blood–brain barrier to protect the brain and reduce the incidence of various brain diseases, such as dementia, Alzheimer disease (AD), Parkinson disease (PD), depression, and ischemia [15,16,17,18,19]. Moreover, cellular studies have demonstrated the cytoprotective and neuroprotective effects of CGA [20,21,22].

The excessive release of glutamate from axon terminals may contribute to neuronal damage and death following a number of neurological insults [3,23], and CGA has been found to protect against glutamate-induced neuronal cell death in experimental models [21,24,25]; however, no studies have investigated the effect of CGA on glutamate release. Therefore, this study examined the effect of CGA on glutamate release from rat cerebrocortical nerve terminals (synaptosomes) and the possible mechanisms underlying this effect. In addition, this study evaluated the neuroprotective effect of CGA in a rat model of kainic acid (KA)-induced glutamate excitotoxicity [26].

## 2. Results

### 2.1. Effect of CGA on 4-Aminopyridine (4-AP)-Evoked Glutamate Release from Rat Cerebrocorticalsynaptosomes

To examine the effect of CGA (Figure 1A) on glutamate release, synaptosomes were stimulated by 4-AP (1 mM), which increases Ca^2+^ influx and glutamate release [27]. As shown in Figure 1B, incubation with CGA (30 µM) for 10 min prior to 4-AP addition did not produce any significant effect on the basal release of glutamate but markedly reduced the 4-AP-induced release of glutamate release in the presence of 1.2 mM CaCl_2_ [t(8) = 11.8, *p* < 0.001 vs. control group]. CGA produced a concentration-dependent inhibition of 4-AP-evoked glutamate release, with a maximum inhibition of 41.4% ± 1.8% produced at 30 µM; the half maximal effective concentration for this inhibition was 20 µM (Figure 1B, inset). In addition, bafilomycin A1, which causes the depletion of glutamate in synaptic vesicles [28], reduced the 4-AP-evoked glutamate release [F(3,16) = 875.1; *p* < 0.001]. When bafilomycin A1 was present, CGA failed to produce significant inhibition (*p* = 0.99 vs. bafilomycin A1-treated group, Figure 1C). By contrast, DL-threo-β-benzyloxyaspartate (DL-TBOA), a nonselective inhibitor of all excitatory amino acid transporter subtypes, increased 4-AP-evoked glutamate release (because of inhibition of reuptake of released glutamate) [29] (F(3,16) = 254.9, *p* < 0.001 vs. control group). When DL-TBOA we present, the inhibitory effect of CGA on 4-AP-evoked glutamate release was not changed (*p* < 0.05 vs. DL-TBOA-treated group; Figure 1C). In addition, the glutamate release evoked by 1 mM 4-AP was reduced in the extracellular-Ca^2+^-free solution that contained 300 µM EGTA (F(3,16) = 396.8, *p* < 0.001 vs. control group). This Ca^2+^-independent release evoked by 4-AP was, however, unaffected by CGA (*p* = 1 vs. Ca^2+^-free group; Figure 1C).

### 2.2. Effect of CGA on 4-AP-Induced Increase in [Ca^2+^]C and Membrane Potential Depolarization in the Synaptosomes

Figure 2A shows that 4-AP elicited a rise in [Ca^2+^]_C_, and CGA prerincubation reduced the 4-AP-induced [Ca^2+^]_C_ increase by 33% (t(8) = 4.456, *p* < 0.001 vs. control group). CGA had no significant effect on the basal [Ca^2+^]_C_ (*p* = 1). In addition, the effect of CGA on synaptosomal plasma membrane potential was determined with the membrane potential-sensitive dye DiSC3(5). 4-AP (1 mM) caused an increase in DiSC3(5) fluorescence. CGA preincubation did not alter the resting membrane potential and produced no significant change in the 4-AP-mediated increase in DiSC3(5) fluorescence [t(8) = 0.959, *p* = 0.4 vs. control group; Figure 2B].

### 2.3. Effect of CGA on Glutamate Release in the Presence of Voltage-Dependent Ca^2+^ Channel Blockers or Intracellular Ca^2+^ Release Inhibitors

Either voltage-gated Ca^2+^ channels (VGCCs) or intracellular Ca^2+^ stores are responsible for the release of glutamate evoked by depolarization [30,31]. In Figure 3, the selective VGCC inhibitors were used to characterize the role of individual Ca^2+^ channel subtypes in the observed CGA-mediated inhibition of glutamate release. With the blockade of N-type VGCCs with ω-conotoxin GVIA (ω-CgTX GVIA, 2 µM), glutamate release evoked by 4-AP was reduced (F(3,16) = 58.1; *p* < 0.001 vs. control group). When ω-CgTX GVIA was present, CGA inhibition of glutamate release was not changed (*p* < 0.001 vs. ω-CgTX GVIA-treated group). Selective blockade of P/Q-type VGCCs using ω-agatoxin IVA (ω-AgTX IVA, 0.5 µM) also reduced control 4-AP-evoked glutamate release (F(3,16) = 50.8; *p* < 0.001 vs. control group). In the presence of ω-AgTX IVA, however, CGA inhibition of glutamate release was markedly abolished (*p* = 0.98 vs. ω-AgTX IVA-treated group). In addition, 4-AP-evoked glutamate release was reduced by dantrolene, an inhibitor of intracellular Ca^2+^ release from the endoplasmic reticulum (ER) (F(3,16) = 305.3, *p* < 0.001 vs. control group), and 7-chloro-5-(2-chlorophenyl)-1,5-dihydro-4,1-benzothiazepin-2(3H)-one (CGP37157), an inhibitor of mitochondrial Na^+^/Ca^2+^ exchange (F(3,16) = 26.9, *p* < 0.001 vs. control group). When dantrolene or CGP37157 was present, CGA significantly reduced 4-AP-evoked glutamate release (*p* < 0.05 vs. dantrolene- or CGP37157-treated group; Figure 3). We also examined the effect of CGA on the 15 mMKCl-evoked glutamate release, a process that involves Ca^2+^ influx primarily through the opening of the VGCCs [32]. As illustrated in Figure 3 inset, CGA also significantly inhibited the KCl-evoked glutamate release [t(8) = 12.6, *p* < 0.001 vs. control].

### 2.4. CGA Interacts with the P/Q-Type Ca^2+^ Channel

To predict the binding effects of CGA with the P/Q-type Ca^2+^ channel protein, the tools of LibDock in Discovery Studio 4.1 client were used for the molecular docking experiment. The crystal structure of P/Q-type Ca^2+^ channel protein (PDB ID 3BXK) was downloaded from RCSB protein data bank. The results of molecular docking showed CGA interacts with amino acid residues ARG 86, ARG 90, and ASP 93 in P/Q-type Ca^2+^ channel protein through hydrogen-bonding interactions (Figure 4).

### 2.5. Effect of CGA on Glutamate Release in the Presence of Calmodulin and Ca^2+^/Calmodulin-Dependent Kinase II (CaMKII) Inhibitors

It has been reported that increased Ca^2+^ influx in nerve terminals enhances CaMKII activation and glutamate release [33]. We next examined whether the inhibition of Ca^2+^ influx caused by CGA decreased CaMKII activity and glutamate release. We tested how 1-[N, O-Bis(5-isoquinolinesulfonyl)-N-methyl-L-tyrosyl]-4-phenylpiperazine (KN62), a general inhibitor of CaMKII, changed the CGA effect. As shown in Figure 5, KN62 reduced the glutamate release induced by 4-AP (F(3,16) = 186.6, *p* < 0.001 vs. control group). When KN62 was present, CGA did not produce any significant inhibition of glutamate release (*p* = 0.4 vs. KN62-treated group). Similar results were also observed in the presence of W7, an inhibitor of calmodulin, which reduced the 4-AP-evoked glutamate release (F(3,16) = 128.1; *p* < 0.001 vs. control group). In addition, CGA did not reduce the release of glutamate evoked by 4-AP when W7 was present (*p* = 0.9 vs. W7-treated group).

### 2.6. Effect of CGA on the Phosphorylation of CaMKII and Its Substrate, Synapsin I, in Cerebrocorticalsynaptosomes

Figure 6 shows that 4-AP increased the phosphorylation of CaMKII (F(2,12) = 238.7; *p* < 0.001 vs. control group) and synapsin I (F(2,12) = 45.1; *p* < 0.001 vs. control group) in the presence of 1.2 mM CaCl_2_. These phosphorylation elevations produced by 4-AP were markedly decreased by the presence of CGA (*p* < 0.05 vs. 4-AP-treated group; Figure 6).

### 2.7. Effect of CGA on the Neuronal Damage and Glutamate Elevation in the Cortex of Rats with KA

We also studied the neuroprotective effect of CGA in a rat model of KA-induced glutamate excitotoxicity. As shown in Figure 7A, rats were intraperitoneal (i.p.) administrated with CGA or dimethylsulfoxide (DMSO) 30 min before KA injection (i.p.) Then, 72 h after the KA injection, neutral red staining of brain sections revealed significant neuronal loss in the frontal cortex of the KA-injected rats compared with that of the DMSO-treated rats (control) (F(3,8) = 36.1, *p* < 0.001). By contrast, neutral red staining in CGA + KA-treated rats demonstrated a significant level of protection against cortical cell death. Neuronal survival in CGA + KA-treated rats was significantly higher than that observed in KA-treated rats (*p* < 0.05; Figure 7B). In addition, the Fluoro-Jade B (FJB) findings were consistent with those of the neutral red staining. FJB-positive cells were observed in the frontal cortex of KA-treated rats (F(3,8) = 122.6, *p* < 0.001 vs. control group; Figure 7A,C), whereas the signal was markedly reduced in CGA + KA-treated rats (*p* < 0.05 vs. KA-treated group). No FJB-positive cells were observed in the control rats. The number of survival and FJB-positive cells in the frontal cortex did not differ between the CGA 10 mg/kg + KA and CGA 50 mg/kg + KA groups (*p* > 0.05). In addition, Figure 7D shows that the glutamate concentration in the cortex at 72 h after KA injection was increased compared with that in the control group (F(3,16) = 7.3, *p* < 0.05). Pretreatment with CGA produced a remarkable decrease in glutamate levels compared to KA group (*p* < 0.05).

## 3. Discussion

Direct modulating glutamate release is crucial because glutamate release from synaptic terminals in the brain is a key mechanism underlying the regulation of excitatory neurotransmission [1]; however, excessive glutamate release has pathological consequences [4]. Thus, the inhibition of glutamate release is crucial to neuroprotection. Numerous biomolecules, including phenolic compounds, with therapeutic effects have been developed as new drug candidates. CGA is a polyphenolic compound such as those commonly found in tea, coffee, and many fruits and vegetables [14]. Studies have reported the effectiveness of CGA in the treatment of neurological conditions such as ischemia, PD, AD, depression, and cognitive deficits [15,16,18,19,34]. However, no data on the effect of CGA on glutamate release are available. This study provides the first published description of the direct effects of CGA on glutamate release from rat synaptosomes. CGA did not affect the basal glutamate release; however, when the synaptosomes were depolarized with 4-AP, CGA inhibited glutamate release, suggesting that CGA reduces the release of glutamate triggered by neuronal activation. This result is consistent with those of electrophysiological studies, which have demonstrated that CGA exerts no significant effects on synaptic transmission or synaptic plasticity under physiological conditions but can mitigate the deterioration of synaptic function in pathological conditions [35].

Glutamate release from synaptosomes is regulated by nerve terminal excitability and Ca^2+^ concentration. For example, the inhibition of Na^+^ channels or the activation of K^+^ channels can stabilize synaptosomal excitability and consequently reduce Ca^2+^ influx and glutamate release [27,36]. In this study, the CGA-induced inhibition of glutamate release from synaptosomes exhibited several notable features. First, when extracellular Ca^2+^ was removed, CGA failed to inhibit the 4-AP-induced release of glutamate, suggesting that the inhibition might be dependent on Ca^2+^ influx. Second, the CGA-induced inhibition of glutamate release was prevented by bafilomycin A1, a vacuolar H^+^ ATPase inhibitor, but not by DL-TBOA, a glutamate transporter inhibitor in the plasma membrane that blocks Ca^2+^-independent glutamate release through transporter reversal [37]. Glutamate release produced by the depolarization of synaptosomes is known to have two components: a physiological relevant Ca^2+^-dependent component, which is produced by the exocytosis of synaptic vesicles containing glutamate, and a Ca^2+^-independent component that results from prolonged depolarization causing a membrane potential-mediated shift of the glutamate transporter steady-state toward the outward direction to effect cytosolic glutamate efflux [29]. Therefore, our observation suggests that a reduction in Ca^2+^-independent glutamate release through the glutamate transporter is not involved in the effect of CGA. Third, CGA counteracted the 4-AP-induced increase in intrasynaptosomal Ca^2+^concentration. This result is consistent with findings that CGA reduced glutamate-induced intracellular Ca^2+^ elevation in cultured cortical neurons [24,25]. Fourth, the Ca^2+^ elevation mediated by 4-AP resulted from both an increase in Ca^2+^ entry through VGCCs and Ca^2+^ release from the ER or mitochondria [30,31]. We observed that CGA did not inhibit glutamate release when P/Q-type Ca^2+^ channels were blocked, and intracellular Ca^2+^ release inhibitors exerted no effect on the action of CGA, suggesting the involvement of a reduction in Ca^2+^ influx mediated by P/Q-type Ca^2+^ channels. In addition, CGA inhibited KCl-induced glutamate release, which involves only Ca^2+^ channels [27]. Furthermore, molecular docking revealed that CGA can form hydrogen bonds with amino acid residues in P/Q-type Ca^2+^ channels. Fifth, CGA exhibited no effect on synaptosomal membrane potential, as measured with DiSC_3_(5) dye. This result indicates that CGA does not reduce synaptosomal excitability, which reduces the influx of Ca^2+^ and thereby reduces glutamate release. Finally, the suppression of calmodulin and CaMKII prevented the inhibition of glutamate release by CGA. Furthermore, CGA suppressed the 4-AP-induced phosphorylation of CaMKII and its substrate, synapsin I. CaMKII in nerve terminals is activated by Ca^2+^ and calmodulinand subsequently phosphorylates synapsin I. This phosphorylation reaction causes cytoskeletal disassembly, which increases the availability of synaptic vesicles and the release of glutamate [38,39,40]. Therefore, CGA directly suppresses P/Q-type Ca^2+^ channels in rats, thereby reducing Ca^2+^-induced activation of CaMKII/synapsin I and subsequently reducing synaptic vesicle availability and glutamate release from rat cerebrocortical synaptosomes.

Additionally, we discovered that CGA exhibited neuroprotection in a rat model of KA-induced excitotoxicity. KA, a glutamate analogue, is a powerful neurotoxin that stimulates the release of glutamate and causes the overstimulation of glutamate receptors [41]. Systemic KA injection damages neurons in several brain regions [42]. The neuronal damage induced by KA resembles that of some neurological diseases; therefore, KA is an ideal tool for investigating the mechanisms underlying neurodegeneration and neuroprotection [26]. In this study, KA significantly increased glutamate levels relative to those of the control group and caused substantial neuronal loss in the cortex, as reported in previous studies [41,43]. These KA-induced alterations were significantly counteracted by CGA pretreatment, indicating that CGA exerts anti-excitotoxic and neuroprotective effects. Our results are in accordance with those of previous studies, which have investigated the neuroprotective effects of CGA by using in vitro and in vivo experimental models and concluded that CGA protects neurons from glutamate-induced death [21,24,25,35]. Because the abnormal release of glutamate leading to neuronal loss is a mechanism underlying numerous neurological diseases [44], inhibiting glutamate release is a key strategy for neuroprotection against excitotoxicity [5,6,7]. Therefore, the ability of CGA to reduce glutamate release from synaptosomes is of particular significance and presents an additional explanation for its protection against glutamate excitotoxicity in addition to its antioxidant and anti-inflammatory actions [18,20]. The relevance of our finding to in vivo clinical situations remains to be determined. In addition, although we observed significant neuroprotective effects of CGA against KA-induced excitotoxic injury, this study has limitations. For example, we studied the effects of CGA on only neurons. Because KA-induced excitotoxicity involves glial activation and because suppression of glial activation prevents the KA-induced neuronal damage [45,46], further investigation is required to elucidate the effects of CGA on the reactions of astrocytes and microglia to KA-induced excitotoxicityin rats. The molecular mechanism underlying the neuroprotective effect of CGA against KA-induced neuronal damage also requires further investigation. On the other hand, previous research had already demonstrated that CGA could increase the level of Gamma-aminobutyric acid (GABA) in pilocarpine-induced epileptic mice brain [47]. Whether CGA can regulate the release of GABA remains to be elucidated.

In summary, numerous biomolecules, including phenolic compounds, exhibit therapeutic potential and have therefore been targeted as new drug candidates. Our data demonstrate that the polyphenolic compound CGA not only inhibits glutamate release from rat cerebrocortical synaptosomes by directly reducing Ca^2+^ entry through P/Q-type Ca^2+^ channels and consequently suppressing the CaMKII/synapsin I pathways but also exhibits neuroprotective effects against KA-induced cortical neuronal injury in rats (Figure 8). Our findings provide valuable new insights into the functions of CGA in the brain and their underlying mechanisms and support the potential use of CGA in the treatment of neuronal diseases that involve glutamatergic abnormalities.

## 4. Materials and Methods

### 4.1. Animals

Experiments were performed using adult male Sprague-Dawley rats (150–200 g). Animals were retained in a pathogen free and temperature-controlled environment with free access to food and water. All experimental protocols were carried out in accordance with the National Institutes of Health Guide for the Care and Use of Laboratory Animals (NIH Publications No. 80-23, revised 1996) and were approved by the Institutional Ethics Committee (A10829), Fu Jen Catholic university. In total, 47 rats were subjected in this study. Fresh brain tissue of 35 rats was used for glutamate release, Ca^2+^ concentration, membrane potential, high-performance liquid chromatography (HPLC), and Western blot. Histological staining was performed on fixed brain tissue of 12 rats.

### 4.2. Materials

CGA was purchased from ChemFaces (Wuhan, Hubei, China) in 98% purity. 4-AP, bafilomycin A1, DL-TBOA, dantrolene, CGP37157, and KN62 were purchased from Tocris Cookson (Bristol, UK). Fura-2-acetoxymethyl ester (Fura-2-AM) was purchased from Invitrogen (Carlsbad, CA, USA). ω-CgTX GVIA and ω-AgTX IVA were purchased from Alomone lab (Jerusalem, Israel). CaMKII, phospho-CaMKII (p-CaMKII), synapsin I, β-actin, and horseradish peroxidase-conjugated secondary antibodies were obtained from Cell Signaling (Beverly, MA, USA). Phospho-synapsin I (Serine 603) (p-synapsin I) antibody was from Genetex (Zeeland, MI, USA). Fluoro-Jade B (FJB) was from Histo-Chem Inc. (Jefferson, AR, USA). KA, DMSO, neutral red and general reagents were from Sigma Chemical Co. (St Louis, MO, USA).

### 4.3. Preparation of Synaptosomes

Synaptosomes were prepared as previously reported [48,49]. Briefly, rats were killed by cervical dislocation and decapitation. The cerebral cortex was rapidly removed and homogenized in ice-cold hepes-buffered medium containing 0.32 M sucrose (pH 7.4). The homogenate was centrifuged at 3000× *g* for 10 min at 4 °C. The supernatant was retained and centrifuged at 14,500× *g* for 12 min at 4 °C. The pellet was resuspended and layered on top of a discontinuous Percoll gradient and centrifuged at 32,500× *g* for 7 min at 4 °C. Protein concentration was determined using the Bradford assay. Synaptosomes were centrifuged in the final wash to obtain synaptosomal pellets with 0.5 mg protein.

### 4.4. Glutamate Release Determination

Synaptosomal pellets were analyzed for glutamate release using enzyme-linked fluorescence method previously described [48]. In brief, synaptosomes were suspended in hepes-buffered medium containing 2 mM NADP^+^, 50 units of glutamate dehydrogenase, and 1.2 mM CaCl_2_, and the synaptosome suspension was stimulated after 5 min with either 1 mM 4-AP or 15 mM KCl. Increases in fluorescence due to production of NADPH was determined (λ excitation = 340 nm and λ emission = 460 nm) with a PerkinElmer LS55 spectrofluorimeter. Released glutamate was calibrated by a standard of exogenous glutamate (5 nmol) and expressed as nanomoles glutamate per milligram synaptosomal protein (nmol/mg). Values quoted in the text and depicted in bar graphs represent the levels of glutamate cumulatively released after 5 min of depolarization, and are expressed as nmol/mg/5 min.

### 4.5. Determination of Cytosolic Free Ca^2+^ Concentration ([Ca^2+^]_C_)

Measurement of [Ca^2+^]_C_ was determined by fluorescent probe fura-2 AM. Synaptosomes were incubated with 5 µM fura-2in the hepes-buffered medium containing 0.1 mM CaCl_2_ for 30 min at 37 °C. After fura-2 loading, the synaptosomes were centrifuged for 1 min at 3000× *g* and pellets were resuspended in the hepes-buffered medium containing 1.2 mM CaCl_2_. Fura-2/Ca fluorescence was monitored in a PerkinElmer LS55 spectrofluorimeter (PerkinElmer Life Sciences, Waltham, MA, USA) at 340 nm and 510 nm. [Ca^2+^]_C_ (nM) was calculated using the equations described previously [50].

### 4.6. Determination of Synaptosomal Membrane Potential

The synaptosomal membrane potential was detected using a positively charged membrane potential-sensitive carbocyanine dye DiSC_3_(5) [51]. DiSC_3_(5) fluorescence was measured (λ excitation = 646 nm and λ emission = 674 nm) with a PerkinElmer LS55 spectrofluorimeter. Data are expressed in fluorescence units.

### 4.7. Molecular Docking Study

The molecular docking experiment was performed by the tools of LibDock in software of Discovery Studio 4.1client (BIOVIA software Inc., San Diego, CA, USA). Briefly, the molecular structure of P/Q-type Ca^2+^ channel protein (PDB ID 3BXK) was downloaded from the RCSB Protein Data Bank, and then prepared following the standard protocols in software. The structure of CGA was drawn and docked into the active site in P/Q-type Ca^2+^ channel protein using the best mode in software.

### 4.8. Western Blot

Western blotting was conducted as previously reported [46]. Briefly, synaptosomes were lysed in an ice-cold Tris–HCl buffer solution, centrifuged for 10 min at 13,000× *g* at 4 °C, and the supernatants were collected. Protein content was determined by using the Bradford assay. Protein (30 µg) was separated by SDS-polyacrylamide gel electrophoresis (10%). After transfer to a nitrocellulose membrane, proteins were detected with a specific primary antibody and a horseradish peroxidase conjugated secondary antibody using enhanced chemiluminescence (ECL, Amersham Biosciences Corp, Amersham, Buckinghamshire, UK). The primary antibodies used in this study include CaMKII (1:10,000), p-CaMKII (1:2000), synapsin I (1:50,000), p-synapsin I (serine 603; 1:2000), and α-actin (1:1000). Quantification was obtained by scanning densitometry of five independent experiments, and analyzed with ImageJ software (Synoptics, Cambridge, UK).

### 4.9. Histological Staining

Rats were divided into four groups: (1) DMSO-treated group (control), (2) KA-treated group, (3) CGA 10 mg/kg + KA-treated group, and (4) CGA 50 mg/kg + KA-treated group. CGA was dissolved in a saline solution containing 1% DMSO and was administered (i.p.) 30 min before KA (15 mg/kg in 0.9% NaCl, pH 7.0, i.p.) injection. Then, 72 h after KA injection, rats were euthanized with CO_2_and perfused transcardially with heparinized saline followed by 4% paraformaldehyde (in 0.1 M phosphate buffer, pH 7.4). The brain was dissected, post fixed for 48 h in the same fixative, and dehydrated using serial dilutions of alcohol. Then the brains were cleared in xylene, embedded in paraffin, and sectioned at a thickness of 20 µm using a microtome. Serial brain sections were mounted on a glass slide and used for neutral red and FJB staining to assess neuronal damage. For neutral red staining, the slides were stained with 1% neutral red for 2 min after dehydration in graded ethanol solutions; they were then rehydrated and coverslipped. FJB staining was used to assess degenerating neurons, as previously described [52]. Briefly, the slides were immersed in 100% ethanol (3 min), 70% ethanol (2 min) and distilled water (2 min). Slides were then incubated in 0.06% potassium permanganate for 15 min, washed twice in distilled water and then immersed in FJB solution (0.001% FJB/0.1% acetic acid) for 30 min in darkness. After washing three times in distilled water, the slides were air dried in the dark for 20 min, dehydrated in xylene, and coverslipped. The neuronal morphology and degeneration of the cortex was observed by an upright fluorescence microscope (Zeiss Axioskop 40, Göttingen, Lower Saxony, Germany). Leica 4× or 10× objective lenses with a numerical aperture (NA) of 0.1 or 0.25 were used in this study (Wetzlar, Germany). The surviving neurons and FJB-positive cells were further counted in an area of 255 μm × 255 μm of the cortex using Image J software (Synoptics, Cambridge, UK).

### 4.10. Determination of Glutamate in Brain Tissue

At 72 h after KA injection, the rats were killed by decapitation and the cortex rapidly dissected. Cortical tissue (50 mg) was homogenized with 200 µL distilled water and centrifuged for 10 min at 1500× *g* (4 °C). The supernatant was filtered through 0.22 µm filters and injected into the HPLC system with electrochemical detection (HTEC-500, Eicom, Kyoto, Japan). The relative free glutamate concentration was determined using peak areas by an external standard method. Serial dilutions of the standards were injected, and their peak areas were determined. A linear standard curve was constructed by plotting peak areas versus corresponding concentrations of each standard [46].

### 4.11. Statistical Analysis

The results were analyzed using SPSS software ((v.24; IBM, Armonk, NY, USA) and expressed as mean ± standard error of the mean (SEM). Comparisons between treatments and control groups were performed by one-way analysis of variance (ANOVA) and post hoc Tukey’s test. A probability of *p* < 0.05 was considered significant.

## Figures and Tables

**Figure 1 ijms-22-11447-f001:**
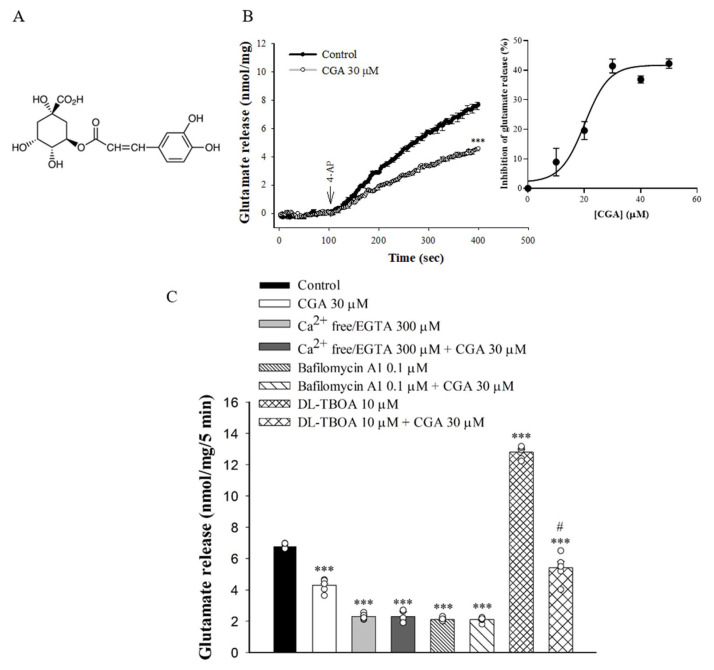
Effect of CGA on 4-AP-evoked glutamate release from rat cerebrocortical synaptosomes. (**A**) The chemical structure of CGA. (**B**) 4-AP-evoked glutamate release from synaptosomes incubated in the presence of 1.2 mM CaCl_2_ and in the absence (control) or presence of CGA. Inset, dose–response curve for CGA inhibition of 4-AP-evoked glutamate release, showing percentage inhibition compared with controls. (**C**) Extracellular Ca^2+^-free solution containing EGTA, glutamate transporter inhibitor DL-TBOA or vacuolar H^+^ ATPase inhibitor bafilomycin A1 on the action of CGA. CGA was added 10 min before the addition of 4-AP, and other drugs were added 10 min before this. Data are the mean ± SEM (*n* = 5 per group). ***, *p* < 0.001 vs. control group. #, *p* < 0.05 vs. DL-TBOA-treated group.

**Figure 2 ijms-22-11447-f002:**
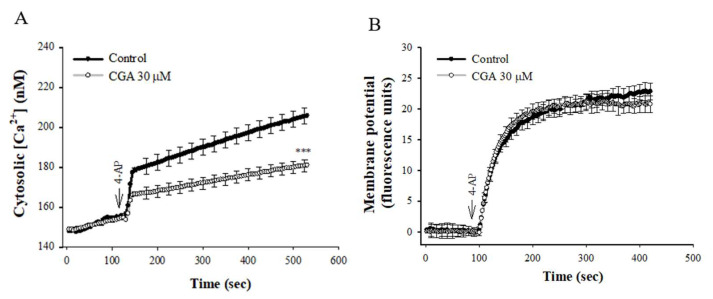
Effect of CGA on [Ca^2+^]_C_ (**A**) and the synaptosomal membrane potential (**B**). CGA was added 10 min before the addition of 4-AP. Data are presented as mean ± SEM. (*n* = 5 per group). ***, *p* < 0.001 vs. control group.

**Figure 3 ijms-22-11447-f003:**
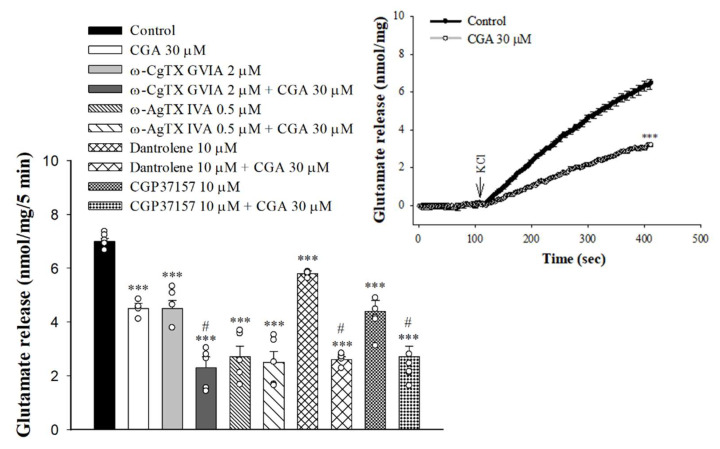
Effect of CGA on 4-AP-evoked glutamate release in the presence of N-type Ca^2+^ channel blocker ω-CgTX GVIA, P/Q-type Ca^2+^ channel blocker ω-AgTX IVA, ryanodine receptor inhibitor dantrolene, or mitochondrial Na^+^/Ca^2+^ exchange inhibitor CGP37157. Inset, effect of CGA on the release of glutamate evoked by 15 mM KCl. CGA was added 10 min before the addition of 4-AP, and other drugs were added 10 min before this. Data are presented as mean ± S.E.M. (*n* = 5 per group). ***, *p* < 0.001 vs. control group. #, *p* < 0.05 vs. ω-CgTX GVIA-, dantrolene- or CGP37157-treated group.

**Figure 4 ijms-22-11447-f004:**
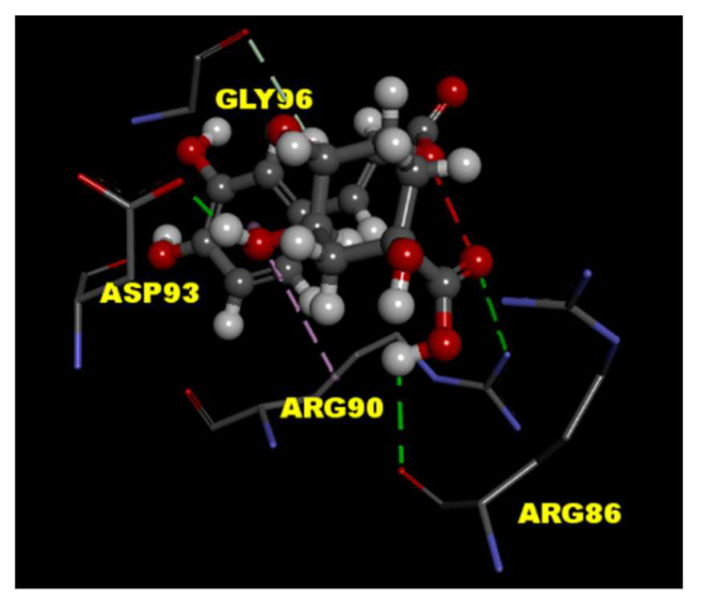
Molecular docking of CGA to P/Q-type Ca^2+^ channel protein. The interaction between CGA with the binding site of molecular structure of P/Q-type Ca^2+^ channel protein (PDB ID 3BXK) was predicted by the Discovery Studio 4.1 software. Hydrogen-bonding interactions (green lines) of protein–ligand as shown.

**Figure 5 ijms-22-11447-f005:**
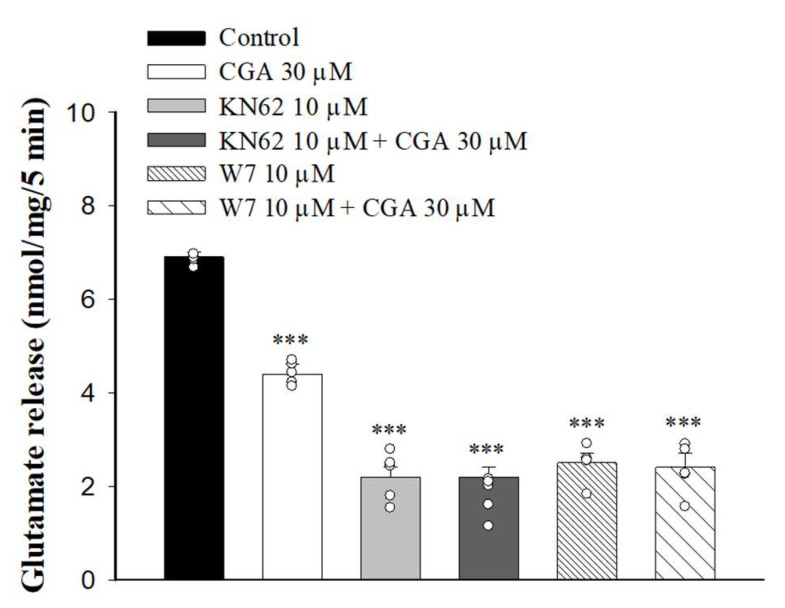
Effect of CGA on 4-AP-evoked glutamate release in the presence of the calmodulin inhibitor W7 or CaMKII inhibitor KN62. CGA was added 10 min before the addition of 4-AP, and other drugs were added 10 min before this. Data are presented as mean ± SEM. (*n* = 5 per group). ***, *p* < 0.001 vs. control group.

**Figure 6 ijms-22-11447-f006:**
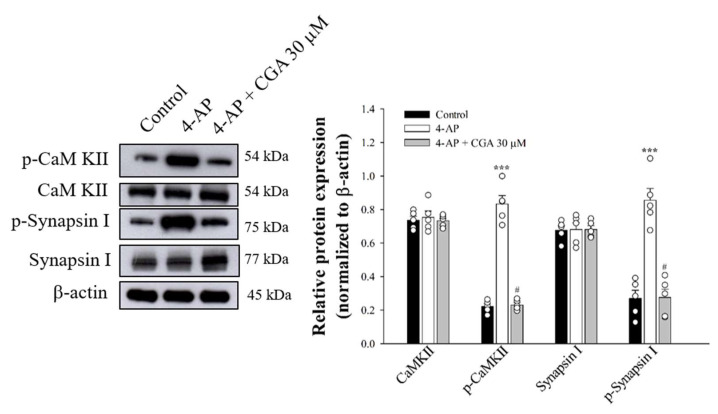
Effect of CGA on CaMKII and synapsin I phosphorylation evoked by 4-AP. CGA was added 10 min before the addition of 4-AP. Data are presented as mean ± SEM. (*n* = 5 per group). ***, *p* < 0.001 vs. control group. #, *p* < 0.05 vs. 4-AP-treated group.

**Figure 7 ijms-22-11447-f007:**
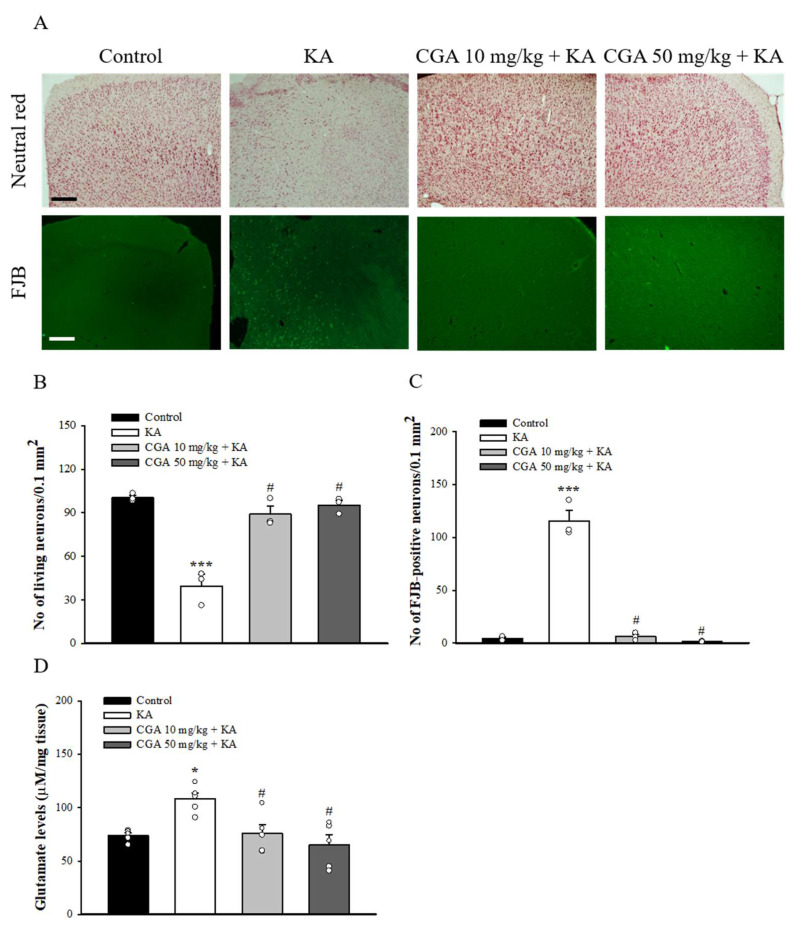
Effect of CGA pretreatment on the neuronal cell death and glutamate levels in the cortex of rats with KA. (**A**) Representative images of neutral red and FJB staining at 3 d after i.p. KA. (**B**, **C**) Quantitative data of A showing the number of living neurons and FJB-positive cells in the cortex. (**D**) The effect of CGA on the concentration of glutamate in the cortex. Data are presented as mean ± SEM. (*n* = 3–5 rats for each group). *, *p* < 0.05, ***, *p* < 0.001 vs. control group. #, *p* < 0.05 vs. KA group. Scale bar: 400 μm.

**Figure 8 ijms-22-11447-f008:**
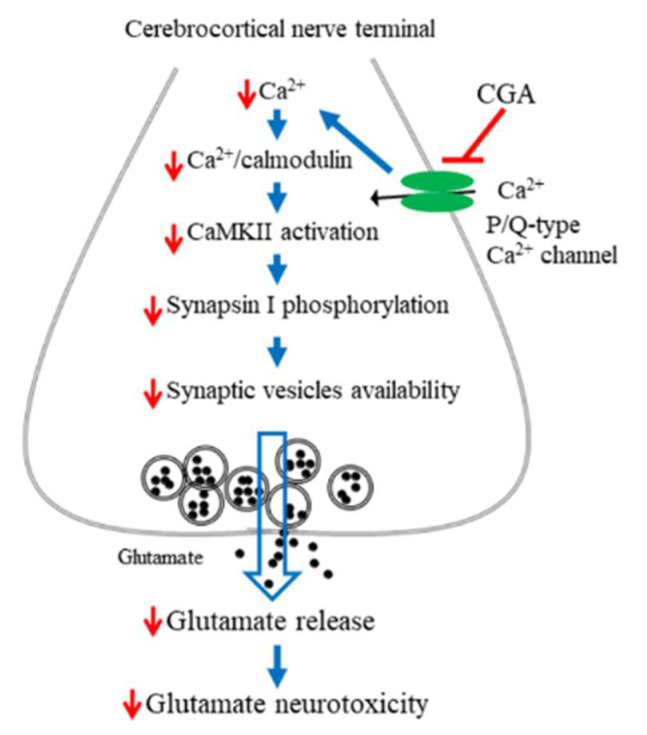
A proposed mechanism underlying the inhibition of glutamate release by CGA. In rat cerebrocortical nerve terminals, CGA direct suppresses P/Q-type Ca^2+^ channels, leading to CaMKII/synapsin I pathway suppression and glutamate release reduction. This effect might contribute to the neuroprotection against KA-induced cortical neuronal damage.

## Data Availability

The data presented in this study are available on request from the corresponding author.

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
