# Peer review of "Chlorogenic Acid Decreases Glutamate Release from Rat Cortical Nerve Terminals by P/Q-Type Ca2+ Channel Suppression: A Possible Neuroprotective Mechanism"

_ijms, 2021, doi:10.3390/ijms222111447_

Round 1

Reviewer 1 Report

The manuscript by Hung YC et al, describes the inhibitory effect of the polyphenolic compound chlorogenic acid (CGA) on glutamate release from rat cortical synaptosomes stimulated with 4-AP and the detailed mechanisms through which this effect takes place in this preparation. Results show that the effects of CGA are due to inhibition of external Ca2+-dependent, vesicular exocytotic release because CGA blocks P/Q type calcium channels, with inhibition of CaMKII/synapsin I pathway. The conclusions from this study are presented in view of neuroprotection against glutamate-induced excitotoxicity. The experimental procedures are well designed and results sound. Functional release studies and other methods give a clear picture of the suggested effects of CGA on glutamate release from nerve endings and the possible implications. As stated by the authors, polyphenols can inhibit the release of glutamate, and several of these compounds including CGA exhibit neuroprotective properties. Although this kind of effects by natural or synthetic compounds in synaptosomes is not a novel finding, the data here presented about CGA and neuroprotection are original and interesting and the manuscripts is a good contribution to this important field.

There are some weak points and concerns that are to be considered.

Also, some considerations , if possible, could ameliorate the study.

General comments:

  1. I might guess, starting from these observations, that in addition to target the glutamatergic system, CGA might be a “more general” inhibitor of neurotransmission, thus including inhibition of release of other transmitters (for instance, GABA release) if its target is the P/Q type calcium channel. Therefore the specificity, and selectivity of effect on glutamatergic transmission is not too clear. This applies perhaps to many other compounds with similar characteristics anyway. This point could limit the relevance of the CGA effect and the therapeutic potential of this and other “similar” compounds. This point should be discussed (if some evidence contraddict these sentences, please report).

  1. In general, compounds able to reduce glutamate release are proposed, and sometimes also used (example: riluzole, that however exhibits limited therapeutic effects) in view of neuroprotection. Several compounds have been found able to suppress glutamate release, even with mechanisms similar to those observed here. In spite of the very widespread and dated use of food/drinks containing CGA, to date, not many clinical studies on the neuroprotective effects of CGA have been performed in humans to the best of my knowledge, and therefore I think the related data are limited. As to the present investigation, it seems there is a mismatch between the very promising evidence from the present study (glutamate release data, strong evidence of neuroprotection against kainic acid-induced neuronal damage) and the limited clinical information available on humans regarding this compound. However this often happens in this kind of studies especially if data on humans are still rather incomplete. The authors could briefly discuss this point, trying if possible to indicate possible partial explanations.

Major specific points:

  1. Results section (Page 2, Para 2.1, line 9 ):

Bafilomycin A1, a vesicular glutamate transporter inhibitor” : this statement is not completely correct. It is actually true that Bafilomycin A1 , as a final effect in the glutamatergic terminals, inhibits uptake of glutamate into synaptic vesicles, but the drug is not directly “a vesicular glutamate transporter inhibitor”. Rather, it is reported to be a selective inhibitor of the vacuolar H+-ATPase which creates the proton gradient “exploited” by processes of neurotransmitter uptake into synaptic vesicles (see for instance: Zhou et al, J Physiol. 2000 May 15; 525(Pt 1): 195–206 , doi: 10.1111/j.1469-7793.2000.t01-1-00195.x) . As a consequence and final effect, Bafilomycin A1 actually does inhibit uptake of neurotransmitters (including glutamate) into synaptic vesicles but this is not due to a direct, selective block of vesicular glutamate transporter.This information on Bafilomycin A1 should be briefly corrected, also in other points of the manuscript, where present (Page 3, legend to Figure 1 lines 5-6; Discussion Section, page 8, line 11).

  1. Results Section (Page 2, Para 2.1, lines 12-20).

 In the absence of external calcium (plus EGTA) or following treatment with bafilomycin, a residual , external Ca2+-independent release remains, as shown in Figure 1 . This release is not sensitive to CGA . Do the author have an idea of how (through which mechanism) this residual glutamate release takes place? It is also stated here that DL-TBOA blocks the Ca2+-independent nonvesicular efflux by transporter reversal ; however, in the presence of DL-TBOA the 4-AP-evoked glutamate release is strongly increased (Fig.1). Also this evident effect of DL-TBOA should be explained in the Discussion Section, considering data already present in the literature if appropriate.

Minor points:

  • Abstract: there is a typo , line 6, the word “and” has to be cancelled
  • Abstract: line 7 , the beginning of the sentence “As observed through molecular docking.... “ is not clear. “As observed” should be better substituted by “In line with evidence observed”....or something similar.
  • Figure 1B (inset). It would be useful if the authors give a brief explanation about how the inhibition percentage is calculated.
  • Page 4, upper part of the legend of Figure 3 : “ω-AgTX VIA” is wrong, it has to be corrected as “ω-AgTX IVA”
  • Page 4, legend of Figure 3 (bottom, last line): “# p<0.05” is also “vs ω-CgTX GVIA-treated group”  
  • Page 9, Figure 8 legend. “This effect might be contributed to.....”: the sentence is not too clear. Maybe “this effect might contribute to.....”.

Author Response

We thank the reviewer for the critical comments and constructive suggestions.

  1. I might guess, starting from these observations, that in addition to target the glutamatergic system, CGA might be a “more general” inhibitor of neurotransmission, thus including inhibition of release of other transmitters (for instance, GABA release) if its target is the P/Q type calcium channel. Therefore the specificity, and selectivity of effect on glutamatergic transmission is not too clear. This applies perhaps to many other compounds with similar characteristics anyway. This point could limit the relevance of the CGA effect and the therapeutic potential of this and other “similar” compounds. This point should be discussed (if some evidence contraddict these sentences, please report).

 According to this point, the sentencesOn the other hand, previous research had already demonstrated that CGA could increase the level of Gamma-aminobutyric acid (GABA) in pilocarpine-induced epileptic mice brain. Whether CGA can regulate the release of GABA remains to be elucidated.are added in the discussion section (Page 10, Lines 22-25)

  1. In general, compounds able to reduce glutamate release are proposed, and sometimes also used (example: riluzole, that however exhibits limited therapeutic effects) in view of neuroprotection. Several compounds have been found able to suppress glutamate release, even with mechanisms similar to those observed here. In spite of the very widespread and dated use of food/drinks containing CGA, to date, not many clinical studies on the neuroprotective effects of CGA have been performed in humans to the best of my knowledge, and therefore I think the related data are limited. As to the present investigation, it seems there is a mismatch between the very promising evidence from the present study (glutamate release data, strong evidence of neuroprotection against kainic acid-induced neuronal damage) and the limited clinical information available on humans regarding this compound. However this often happens in this kind of studies especially if data on humans are still rather incomplete. The authors could briefly discuss this point, trying if possible to indicate possible partial explanations.

 As suggestion by the reviewer, several sentences are modified to Therefore, the ability of CGA to reduce glutamate release from synaptosomes is of particular significance andpresents an additional explanation for its protection against glutamate excitotoxicity in addition to its antioxidant and anti-inflammatory actions[18,20]. The relevance of our finding to in vivo clinical situations remains to be determined.(Page 10, Lines 10-14).

Major specific points:

  1. Results section (Page 2, Para 2.1, line 9 ):

Bafilomycin A1, a vesicular glutamate transporter inhibitor” : this statement is not completely correct. It is actually true that Bafilomycin A1 , as a final effect in the glutamatergic terminals, inhibits uptake of glutamate into synaptic vesicles, but the drug is not directly “a vesicular glutamate transporter inhibitor”. Rather, it is reported to be a selective inhibitor of the vacuolar H+-ATPase which creates the proton gradient “exploited” by processes of neurotransmitter uptake into synaptic vesicles (see for instance: Zhou et al, J Physiol. 2000 May 15; 525(Pt 1): 195–206 , doi: 10.1111/j.1469-7793.2000.t01-1-00195.x) . As a consequence and final effect, Bafilomycin A1 actually does inhibit uptake of neurotransmitters (including glutamate) into synaptic vesicles but this is not due to a direct, selective block of vesicular glutamate transporter.This information on Bafilomycin A1 should be briefly corrected, also in other points of the manuscript, where present (Page 3, legend to Figure 1 lines 5-6; Discussion Section, page 8, line 11).

 As suggestion by the reviewer, the sentence is changed to bafilomycin A1, which causes the depletion of glutamate in synaptic vesicles (Page 2, Lines 32-22); vacuolar H+ ATPase inhibitor bafilomycin A1 (Page 3, Line 6); bafilomycin A1, a vacuolar H+ ATPase inhibitor (Page 9, Line 16)

  1. Results Section (Page 2, Para 2.1, lines 12-20).

 In the absence of external calcium (plus EGTA) or following treatment with bafilomycin, a residual , external Ca2+-independent release remains, as shown in Figure 1 . This release is not sensitive to CGA . Do the author have an idea of how (through which mechanism) this residual glutamate release takes place? It is also stated here that DL-TBOA blocks the Ca2+-independent nonvesicular efflux by transporter reversal ; however, in the presence of DL-TBOA the 4-AP-evoked glutamate release is strongly increased (Fig.1). Also this evident effect of DL-TBOA should be explained in the Discussion Section, considering data already present in the literature if appropriate.

  According to this point, the sentence is modified toDL-threo-β-benzyloxyaspartate (DL-TBOA), a nonselective inhibitor of all excitatory amino acid transporter subtypes, increased 4-AP-evoked glutamate release (because of inhibition of reuptake of released glutamate)(Page 2, Lines 36-38). In addition, in the discussion, several sentences Glutamate release produced by the depolarization of synaptosomes is known to have two components: a physiological relevant Ca2+-dependent component, which is produced by the exocytosis of sunaptic vesicles containing glutamate, and a Ca2+-independent component that results from prolonged depolarization causeing a membrane potential-mediated shift of the glutamate transporter steady-state toward the outward direction to effect cytosolic glutamate effluxare added (Page, Lines 19-25).

Minor points:

  • Abstract: there is a typo , line 6, the word “and” has to be cancelled

The word is deleted (Page 1, Line 6).

  • Abstract: line 7 , the beginning of the sentence “As observed through molecular docking.... “ is not clear. “As observed” should be better substituted by “In line with evidence observed”....or something similar.

The sentence is changed to In line with evidence observed(Page 1, Line 7).

  • Figure 1B (inset). It would be useful if the authors give a brief explanation about how the inhibition percentage is calculated.

In order to more clear, the sentence is modified to dose-response curve for CGA inhibition of 4-AP-evoked glutamate release, showing percentage inhibition compared with controls.(Page 3, Lines 4-5). 

  • Page 4, upper part of the legend of Figure 3 : “ω-AgTX VIA” is wrong, it has to be corrected as “ω-AgTX IVA”

The word is corrected.

  • Page 4, legend of Figure 3 (bottom, last line): “# p<0.05” is also “vs ω-CgTX GVIA-treated group”  

The sentence is modified to p < 0.05 vs ω-CgTX GVIA-, dantrolene- or CGP37157-treated group(Page 5, Line 6).

  • Page 9, Figure 8 legend. “This effect might be contributed to.....”: the sentence is not too clear. Maybe “this effect might contribute to.....”.

The sentence is modified to This effect might contribute to the neuroprotection against KA-induced cortical neuronal damage(Page 10, Lines 36-37).

Reviewer 2 Report

Although very interesting and generally well designed, the reviewed manuscript contains minor issues that needs to be clarified before the publication.

Comments:

  1. More detailed description of animal euthanasia must be provided
  2. It is not clear do the authors tested the specificities of primary antibodies used
  3. FJB staining images included in figure 7 are illegible. Please provide pictures of better quality

Author Response

We thank the reviewer for the critical comments and constructive suggestions.

Reviewer 2

  1. More detailed description of animal euthanasia must be provided

According to this point, the sentencerats were killed by cervical dislocation and decapitationis included in the method section (Page 11, Lines 25-26).

  2. It is not clear do the authors tested the specificities of primary antibodies used

The sentence is modified to The primary antibodies used in this study include CaMKII(1:10000), p-CaMKII (1:2000), synapsin I (1: 50000),p-synapsin I (serine 603; 1: 2000), andb-actin (1: 1000). Quantification was obtained by scanning densitometry of five independent experiments, and analyzed withImageJ software(Page 12, Lines 24-17).

  3. FJB staining images included in figure 7 are illegible. Please provide pictures of better quality

As suggestion by the reviewer, the quality of FJB staining images (Figure 7) is improved. 
